# Structural Characterization and Anti-Inflammatory Effects of 24-Methylcholesta-5(6), 22-Diene-3β-ol from the Cultured Marine Diatom *Phaeodactylum tricornutum*; Attenuate Inflammatory Signaling Pathways

**DOI:** 10.3390/md21040231

**Published:** 2023-04-03

**Authors:** Kalpa W. Samarakoon, Anchala I. Kuruppu, Ju-Young Ko, Ji-Hyeok Lee, You-Jin Jeon

**Affiliations:** 1Institute for Combinatorial Advanced Research and Education (KDU-CARE), General Sir John Kotelawala Defence University, Ratmalana 10390, Sri Lanka; 2Department of Marine Life Sciences, Jeju National University, Jeju 690-756, Republic of Korea; 3Research Institute of Basic Sciences, Incheon National University, Incheon 406-772, Republic of Korea; 4Lee Gill Ya Cancer and Diabetes Institute, Incheon 406-840, Republic of Korea

**Keywords:** *Phaeodactylum tricornutum*, cultured marine diatom, zebrafish, anti-inflammatory effects

## Abstract

In the present investigation, 24-methylcholesta-5(6), 22-diene-3β-ol (MCDO), a major phytosterol was isolated from the cultured marine diatom, *Phaeodactylum tricornutum* Bohlin, and in vitro and in vivo anti-inflammatory effects were determined. MCDO demonstrated very potent dose-dependent inhibitory effects on the production of nitric oxide (NO) and prostaglandin E_2_ (PGE_2_) against lipopolysaccharide (LPS)-induced RAW 264.7 cells with minimal cytotoxic effects. MCDO also demonstrated a strong and significant suppression of pro-inflammatory cytokines of interleukin-1β (IL-1β) production, but no substantial inhibitory effects were observed on the production of cytokines, including tumor necrosis factor-α (TNF-α) and interleukin-6 (IL-6) at the tested concentrations against LPS treatment on RAW macrophages. Western blot assay confirmed the suppression of inducible nitric oxide synthase (iNOS) and cyclooxygenase-2 (COX-2) protein expressions against LPS-stimulated RAW 264.7 cells. In addition, MCDO was assessed for in vivo anti-inflammatory effects using the zebrafish model. MCDO acted as a potent inhibitor for reactive oxygen species (ROS) and NO levels with a protective effect against the oxidative stress induced by LPS in inflammatory zebrafish embryos. Collectively, MCDO isolated from the cultured marine diatom *P. tricornutum* exhibited profound anti-inflammatory effects both in vitro and in vivo, suggesting that this major sterol might be a potential treatment for inflammatory diseases.

## 1. Introduction

Inflammation is an important physiological feature of the host to maintain good health and protection against bacterial or viral infections and tissue injury. Adverse or excessive inflammatory conditions contribute to acute or chronic conditions and lead to many diseases, such as cardiovascular diseases, diabetes, and cancers [1]. Several immune cells, for instance, macrophages, neutrophils, and lymphocytes, are activated in response to inflammation. A respiratory burst of inflammatory cells during inflammation increases the production of reactive oxygen species (ROS), nitric oxide (NO), and prostaglandin E_2_ (PGE_2_), as well as pro-inflammatory cytokines, such as interleukin (IL)-1β, IL-6, and tumor necrosis factor (TNF)-α, which is formed by the activation of inducible nitric oxide synthase (iNOS) and cyclooxygenase-2 (COX-2) [2]. Endotoxins, such as lipopolysaccharides (LPS), stimulate RAW cells by enhancing the NO concentration in this context. Thus, in the treatment of inflammatory diseases, it is important to inhibit the production of these inflammatory mediators.

Marine diatom is a rich component of the phytoplankton and constitutes a major nutritional source in aquaculture. It is recognized as the most vital photosynthetic eukaryote in marine ecosystems [3]. In general, per year, marine phytoplankton produces around 5 × 10^16^ g of organic carbon in the deep sea where lipids are one of the key fractions of phytoplankton biomass. In terms of lipids, dietary polyunsaturated fatty acids (PUFAs) and sterols are found to be major lipid component requirements in marine nutrition [4]. However, phytosterol plays a significant role, and the concentration of phytosterol depends on the physiological state of the microalgae population [5]. Different microalgae comprise various types of sterols, which characteristically vary in each species. Moreover, fatty acids are the potential biomarkers of microalgae [6]. Diatom biomass plays a vital role in other phytoplankton by controlling reproduction and growth in marine animals [7].

*Phaeodactylim tricornutum*, a marine pinnate diatom of Bacillariophyceae (class diatomphyceae), has been known as an important species among marine microalgae. Therefore, *P. tricornutum* diatom has been extensively studied for ecology [8], marine pollution [9], phytosterol [10], biological applications [11,12], and structural diversity [13]. It has been found to have large amounts of functional ingredients, such as pigments [14], lipids [15,16,17], sulfated polysaccharides [11], crude polysaccharides [18], proteins/enzymes [19], peptides [20,21], and antioxidants [22]. In addition, *P. tricornutum* is considered a major source of foods for the commercial rearing of aquaculture invertebrates, and as such, culturing marine diatom has been extensively implemented.

The ocean represents a huge untapped resource for the discovery of new chemicals with pharmaceutical properties. Throughout the globe, determining pharmacologically active metabolites from the ocean has amplified. In this regard, finding novel bioactive natural products from marine bio-entities has shown to be promising. Marine bio-entities comprise functional ingredients, which are quite species-specific, and secondary metabolites, which may be used to defend against infections, predation, homeostasis, and parasitism [23]. The secondary metabolites of marine species, including microalgae, have shown a huge number of medicinal properties. Thus, after being cultured, microalgae have been largely used as pharmaceutically active functional elements. It is also noteworthy that a lot of attention has been given lately to the possibility of replacing compounds of a synthetic nature with natural sources [24].

Amongst the marine diatom lipids, high contents of PUFAs have been reported, around 30–45%. Further, eicosapentaenoic acid (EPA) is one of the major components, making up around 20–40% of cultured *P. tricornutum* [12,16]. However, phytosterol takes a significant place among marine lipids and the concentration of phytosterol depends on the physiological state of the microalgae population [5]. Various microalgae encompass different types of sterols, which characteristically vary in each species. Furthermore, 24-methylcholesta-5(6), 22-diene-3β-ol (brassicasterol) was quantified and extracted from *P. tricornutum* as the major sterol [25] around 40 years ago, but its biological activities have not been assessed or published extensively. Therefore, the objective of this study was to determine the possible anti-inflammatory effect of the isolated brassicasterol from the cultured marine diatom *P. tricornutum* using in vitro and in vivo techniques.

## 2. Results

### 2.1. Structure Elucidation of the Compound 24-Methylcholesta-5(6), 22-Diene-3β-ol from the Cultured Marine Diatom Phaeodactylum tricornutum

The in vitro biological activity guided fractionation of cultured marine diatom *P. tricornutum* hexane fraction (550 mg) was fractionated using solid–liquid phase chromatography (normal-phase silica) column (3 cm × 22 cm) with increasing hydrophilic character using hexane and ethyl acetate solvents. A total of 14 fractions were identified after pooling the elutions followed by thin-layer chromatography (TLC) studies. Fraction 1 (Com 1; 10.4 mg) and fraction 7 (Com 2; 19 mg) were separated with enough purity in the following solvent conditions: hexane-ethyl acetate (95:5, *v*/*v*) and (90:10, *v*/*v*), respectively (Figure 1). In addition, fraction 8 (F8; 90 mg) was further fractionated using preparative thin layer chromatography (PTLC) into three sub-fractions including fraction 8-1 (Com 3; 16.8 mg), fraction 8-2 (Com 4; 10.0 mg) and fraction 8-3 (Com 5; 6.3 mg) respectively. The purity of each of the fractions was enough to elucidate the structures, followed by mass spectroscopy analysis.

The molecular formula, abbreviations, and IUPAC names of the isolated compounds are listed in Table 1.

Among the isolated five different compounds from the cultured microalgae *P. tricornutum*, the yield of MCDO was 1.15%. The molecular mass and molecular formula were determined to be C_20_H_38_O in the ESI (negative mode) as the [M-H] peak at 397.14 *m*/*z*. The calculated molecular mass (398.14 *m*/*z*) was examined with six degrees of unsaturation. The ^13^C-NMR spectrum demonstrated 28 signals, one for each carbon atom of the molecule. Out of these, four signals appeared above 100 ppm, and the signals at δ 140.73, δ 121.71, δ 136.63, and δ 131.64 ppm were assigned to two olefinic bonds. The ^13^C-NMR signals at δ 71.9 ppm revealed the presence of one oxygenated methine signal, which was most likely to be hydroxymethine carbon. Heteronuclear correlation (HSQC) spectrums demonstrated the existence of OH substitution at the C-3 position. Examination of the heteronuclear correlations in the HSQC spectrum demonstrated that the signals of three methyl groups were at δ 20.24 (C-26), 19.7 (C-27), and 18.78 ppm (C-28). Moreover, the olefinic bond at C-22 and C-23 with δ 136.63 and δ 131.64 ppm was confirmed in the tail aliphatic chain. Heteronuclear multiple bond correlation (HMBC) verified the olefinic bond coherence to the C-5 and C-6 positions, at δ 140.73 and 121.71, respectively. The HMBCs are shown in Table 2. The spectrum showed ^13^C-NMR signals due to two unsaturated olefinic groups, and the molecule was assumed to be tetracyclic to account for the six degrees of unsaturation. Additional insight into the molecular structure came from the ^1^H-NMR spectrum, which showed the presence of six methyl groups. This suggests that the molecule has a sterol character. Furthermore, Table 2 shows the assigned multiplicity of the isolated compound. The NMR data, which comply with the current literature, indicate that the isolated compound had the same skeleton as brassicasterol [26,27]. Therefore, the isolated compound (Com 5) is confirmed to be 24-methylcholesta-5(6), 22-diene-3β-ol (MCDO) from the cultured marine diatom *P. tricornutum*.

### 2.2. Inhibitory Effect of MCDO on LPS-Induced Nitric Oxide Production and Cytotoxicity

It has been shown that nitric oxide (NO) is a signaling molecule that plays a main part in inflammation. NO is stimulated by iNOS, which is an inducible isoform. Different concentrations of MCDO were isolated from the cultured marine diatom *P. tricornutum,* and the effects of NO production were determined in LPS-activated RAW 264.7 cells. Inflammation is caused by LPS-stimulated cells, which form NO molecules. The development of the NO signaling concentration was suppressed in the medium, which was followed by the incubation of various doses of MCDO (Figure 2A). Moreover, the MCDO did not demonstrate a cytotoxicity effect against the RAW 264.7 cells at each of the concentrations tested (Figure 2B). The LDH release assay illustrated very low or no cytotoxicity on macrophages at all tested concentrations. Hence, MCDO is marked as a probable agent for attenuating NO production without cytotoxicity effects.

### 2.3. Inhibitory Effect of MCDO on LPS-Induced PGE_2_ Production

It was observed that MCDO inhibited LPS-induced PGE_2_ production in a dose-dependent manner. The suppression of PGE_2_ production was 63% at a 25 μg mL^−1^ concentration, whereas 100% was suppressed with the MCDO treatment. Therefore, MCDO may induce an anti-inflammatory effect by potently inhibiting the production of PGE_2_ when incubated with RAW macrophages (Figure 3A).

### 2.4. Inhibitory Effect of MCDO on LPS-Induced Pro-Inflammatory Cytokine (TNF-α, IL-1β, and IL-6) Production

The stimulation and inhibition of different cytokines are intertwined in a complex pathway that regulates inflammation. At different points of the inflammatory pathway, inflammatory stimulators, such as LPS, induce cytokines that process the activation of macrophages and facilitate tissue responses. The inhibitory effect of MCDO on the development of pro-inflammatory cytokines, such as TNF-α, IL-1β, and IL-6, is usually seen in LPS-stimulated RAW 264.7 macrophages. At the tested concentrations, the MCDO did not demonstrate significant inhibitory activities on the production of cytokines such as TNF-α and IL-6 (Figure 3B,D). However, potent suppression of the production of cytokine IL-1β was seen by the pretreatment of MCDO with RAW macrophages. Hence, the inhibition pathway may be facilitated by the association of IL-1β formation (Figure 3C).

### 2.5. Inhibitory Effect of MCDO on LPS-Induced iNOS and COX-2 Protein Expression

In order to determine the protein expressions of the iNOS and COX-2 proteins, which facilitate the suppression effect of MCDO on NO and PGE_2_ production, Western blots were carried out (Figure 4).

It was observed that the iNOS and COX-2 protein expressions were remarkably stimulated when the macrophages were treated with LPS compared to the control. Nevertheless, when the cells were incubated with MCDO, which was followed by LPS treatment, both iNOS and COX-2 protein expressions were observed to be downregulated, and at the concentration of 25 μg mL^−1^, COX-2 protein expression was entirely blocked. Collectively, this implies that the potent inhibitory effect of PGE_2_ production with the effect of MCDO was associated with the downregulation pattern of the COX-2 protein expression. Further to this, MCDO inhibited the production of NO levels by suppressing the iNOS proteins.

### 2.6. Evaluation of Toxicity and Survival Rate of MCDO in the Zebrafish Embryo

Further studies were conducted to confirm the anti-inflammatory effect of the isolated compound. Thus, MCDO was assessed on the zebrafish model and used in an in vivo assay. For this study, the rates of survival of the zebrafish embryos were monitored with the effects of LPS-induced oxidative stress and MCDO treatments at 12.5, 25, and 50 μg·mL^−1^ concentrations, as shown in Figure 5A. According to the results, the MCDO demonstrated a protective effect on the zebrafish embryos without dose dependency. Hence, the rate of survival of the zebrafish embryos was 100% the same as the control at all of the MCDO-tested concentrations. Therefore, the determined in vitro anti-inflammatory activity of MCDO could be used further to carry out in vivo tests without toxicity effects at the tested concentrations in the zebrafish model.

### 2.7. Inhibitory Effect of the MCDO on LPS-Induced ROS Generation in Zebrafish

The effect of the scavenging of MCDO against LPS-induced oxidative stress in zebrafish was examined via ROS staining with DCF-DA (Figure 5B). As illustrated, the positive control demonstrated lower expression (Figure 5B-I non-treated group), and the LPS-treated negative control (Figure 5B-II LPS-treated group) demonstrated high expression, illustrating oxidative damage. When treated with MCDO, the intracellular ROS was scavenged, and the zebrafish embryos were preserved from oxidative stress-related damage. This was demonstrated significantly in a dose-dependent manner by the treatment concentrations of 12.5, 25, and 50 μg mL^−1^ of MCDO (Figure 5 II–V). ROS generation increased by 240% following the LPS treatment, decreased significantly with the sample treatments, and decreased by 100% at the 50 μg mL^−1^ concentration. These results illustrate that MCDO could lower oxidative stress and that it might scavenge ROS, protecting against LPS-induced oxidative stress.

### 2.8. Inhibitory Effect of MCDO on LPS-Induced Nitric Oxide (NO) Production in Zebrafish

In this experiment, the effect of the inhibition of MCDO on LPS-induced NO production in zebrafish embryos was evaluated. Hence, to analyze the NO production, a fluorescent probe dye, diaminofluorophore 4-amino-5-methylamino-2′,7′-dichlorofluorescein diacetate (DAF-FM DA), was used. The effects of the inhibitory activity of NO production displayed with the treatment of MCDO (Figure 6). In line with the results, treated MCDO demonstrated a significant NO production inhibitory effect. Interestingly, a complete NO production inhibitory effect was reached at the lowest concentration, which was 12.5 μg mL^−1^ of MCDO compared to the positive control of LPS-induced oxidative stress (Figure 6 II–V).

### 2.9. Protective Effect of MCDO on LPS-Induced Cell Death in Live Zebrafish

The induction of cell death by LPS treatment was measured via propidium iodide (PI) by the fluorescence intensity in the body of the zebrafish. As illustrated in Figure 7, cell death was lowered by the introduction of MCDO into the zebrafish that were exposed to LPS (Figure 7A). The microscopic pictures represented in Figure 7A show that the control in the zebrafish model had intact nuclei and that LPS-treated zebrafish displayed a significant increase in the intensity of PI. Nevertheless, in the treated samples, a dramatic reduction in cell death was observed.

## 3. Discussion

*P. tricornutum* is a diatom that has been widely studied in the field of biotechnology. It is a well-known marine diatom that collects different unsaturated fatty acids. The compositions of fatty acids related to *P. tricornutum* biomasses have different functional values that are significant for the progress of marine biotechnology [28]. The two most common methods of culturing marine microalgae systems are either closed (photobioreactor) or open culture systems that are pertinent to the energy source and type of culture. However, the culturing systems may have either favorable or unfavorable effects, but the photosynthetic organisms and their mass production might directly be associated with the source of energy [29]. The physicochemical condition with the operational inputs, such as salt, dissolved CO_2_, water, nutrients, pH, and O_2_, must supply an optimum condition for a steady environment without being adulterated by photo-bioreactors [30]. It is considered that the key ingredients available in cultured marine microalgae could be a good source for the functional food industry [31,32].

Plant-derived sterols (phytosterols), as unsaturated forms, are more present in nature than saturated forms (phytostanols). Phytosterol treatments on a murine experimental model of colitis showed that elevating the significant antioxidant effects reduces plasmatic cholesterol levels and increases gastrointestinal anti-inflammatory effects [33].

The results of previous studies on microalgae culturing and harvesting of the marine diatom *P. tricornutum* biomass have been discussed [34]. According to a previous study, biological activity-guided fractionation resulted in the isolation of five diverse compounds from the hexane fraction of the cultured marine diatom *P. tricornutum*. The isolated fatty alcohol ester, nonyl 8-acetoxy-6-methyloctanoate (NAMO), has been examined for inhibiting the growth of human leukemia cancer cells significantly through the p53 and caspase-3 mediated cell apoptotic pathway [35]. In the present study, the isolated compound 24-methylcholesta-5(6), 22-diene-3β-ol (MCDO) from the hexane fraction of the same cultured *P. tricornutum* was evaluated for its anti-inflammatory effects in vitro and in vivo. According to the literature, MCDO is known as Brassicasterol, a natural product belonging to phytosterols [36], which is the main compound present in canola oil and rapeseed [37]. Further, recent studies of brassicasterol have revealed its anti-infective actions against HSV-1 and *Mycobacterium tuberculosis*, along with cardiovascular protective effects in terms of ACE inhibitory activity and the nutritional and biological values which were originally claimed [38].

Nitric oxide (NO) is an intracellular messenger that facilitates diverse signaling pathways in target cells and plays a key role in many cellular functions in the cardiovascular, nervous, and immune systems [39]. The acute expression of NO assembly is vital in protecting organs, such as the liver, from ischemic damage. Nevertheless, the chronic expression of NO is correlated with many inflammatory conditions [40]. Moreover, in pathological conditions, iNOS increases NO production [41,42]. Nevertheless, the probable therapeutic value associated with the inflammation and inhibition of NO production by the suppression of iNOS expression can be considered. The intense inhibitory effect on NO production has been shown by MCDO, and it was further assessed for its ability to inhibit LPS-induced PGE2 production in RAW 264.7 macrophages. As reported by Nakagawa [43], PGE2 is the most abundant prostanoid in humans, and it is involved in facilitating many various fundamental biological functions. The induction of COX-2 activity and the subsequent generation of PGE2 is similarly related to NO production [44]. The inhibition of iNOS enzyme activity and NO production has been shown to block prostaglandin release from RAW264.7 macrophages [45]. In addition, cytokines are the chemical messengers among the immune cells that play a major role in facilitating inflammatory and immune responses. Pro-inflammatory cytokines, including IL-1β, IL-6, and TNF-α, are largely formed by activated macrophages that activate and also boost the production of PGE_2_ in various cell types [46]. The insightful ability of MCDO to suppress reactive oxygen species (ROS) might be correlated with the inhibition of iNOS and COX-2 protein expression, as well as the inhibition of IL-1β and IL-6 pro-inflammatory cytokines; therefore, MCDO has shown anti-inflammatory potential.

In the recent past, zebrafish (*Daniorerio*) have been widely used in fields, such as cancer and stem cell research. This small tropical freshwater fish has been demonstrated to be a beneficial vertebrate model organism due to its small size, large clutches, low-cost maintenance, and physiological and morphological resemblance to mammals [47]. Further, this transparent larva is easy to rear, and its innate immune system is available until several weeks post-fertilization. LPS, or endotoxins, induce potent inflammation and tolerance effects in the zebrafish model, which have been studied. Several studies have shown that LPS induced toxic effects of phenotypic changes on zebrafish embryos, including the size of the yolk sac edema, varying heart rates, and tail bending. In addition, there were increasing effects of ROS and NO with abnormal cell death, which was also induced by the LPS-stimulated inflammatory responses in zebrafish larvae [48]. Pro-inflammatory cytokines, such as IL-1β, IL-6, and TNF-α, lipid mediators, and ROS are produced in terms of the response of LPS stimulation at an early stage of larvae (2 dpf) [49]. Therefore, they are a sophisticated tool for immune response. Due to these characteristics of the zebrafish model, it is useful for a variety of inflammatory studies and drug discovery and toxicological studies [50]. In this study, the survival rates of zebrafish embryos with the effects of LPS-induced oxidative stress were protected by dose-independent treatment with MCDO. Intracellular ROS was scavenged when the treatment of MCDO and the zebrafish embryos were protected from oxidative stress-related damage. This was demonstrated significantly in a dose-dependent manner. By using the oxidation-sensitive dye DCF-DA as the substrate, the generation of intracellular ROS was detected. The DCF-DA displayed no fluorescence without ROS and became fluorescent upon dealing with ROS. This technique is widely used as a fluorescent probe to assess oxidative activity in cells. Zebrafish significantly increased ROS generation by 240% following the LPS treatment. However, the isolated compound, MCDO from *P. tricornutum*, reduced the level of ROS in zebrafish at the level of control of a 50 μg mL^−1^ concentration. Moreover, an attenuating effect of MCDO on the LPS-induced NO production in zebrafish embryos was detected. In addition to that, the cell death induced by LPS treatment was recovered by MCDO, suggesting that the LPS induced inflammation completely. The fluorescent probe dye DAF-FM DA was used in this assay, and the transformation of DAF-FM DA by NO in the presence of dioxygen generated so-called highly fluorescent triazole derivatives. The LPS significantly elevated the levels of NO in the zebrafish compared to the negative control (zebrafish without LPS treatment). Nevertheless, the NO levels were decreased significantly by the effects of MCDO. In the zebrafish study, there was no significant cell death observed. As such, this suggests that the embryos were not toxic for the isolated sterol compound from the diatom. In many studies, the zebrafish model has been proven to be a reliable technique, and it is accepted to be an efficient and effective anti-inflammatory assay [51,52]. Collectively, the above results imply that MCDO can act as a strong inhibitor of ROS and can induce NO, based on the LPS-stimulated inflammatory zebrafish model. Furthermore, MCDO had a protective effect against the toxicity induced by LPS exposure in the zebrafish embryos. This outcome could explain the probable anti-inflammatory activity of MCDO, which may have a valuable effect during the treatment of inflammatory diseases.

## 4. Materials and Methods

### 4.1. Chemicals and Reagents

Dulbecco’s Modified Eagle’s Medium (DMEM), fetal bovine serum (FBS), and penicillin–streptomycin were obtained from Gibco/BRL (Burlington, ON, Canada). 3-(4, 5-Dimethylthiazol-2-yl)-2, 5-diphenyltetrazolium bromide (MTT), propidium iodide (PI), 2,7-dichloro-dihydro fluorescein diacetate (DCFH-DA), diaminofluorophore 4-amino-5-methylamino-2′,7′-dichlorofluorescein diacetate (DAF-FM DA), and dimethyl sulfoxide (DMSO) were obtained from Sigma–Aldrich (St. Louis, MO, USA). All reagents and chemicals were of analytical grade and used as received without further purification.

### 4.2. Extraction, Isolation, and Characterization of Compounds

*P. tricornutum* diatom (38 g), which was lyophilized and homogenized, was subjected to extraction three times with (80%) methanol in each 90 min sonication period at 25 °C. The crude methanol extract that was obtained was further exposed to solvent–solvent partition chromatography. Subsequently, four different fractions, including hexane, chloroform, ethyl acetate, and aqueous extracts, were separated with varying polarity.

Based on the sequential chromatographic purification steps described by Samarakoon et al. 2014 [35], 5 different structural formulas were elucidated and are presented in Figure 1.

### 4.3. Cell Culture Method

RAW 264.7, a murine macrophage cell line was purchased from the Korean Cell Line Bank (KCLB, Seoul, Republic of Korea). RAW 264.7 cells were maintained in DMEM supplemented with 10% FBS, 100 U mL^−1^ of penicillin, and 100 μg mL^−1^ of streptomycin at 37 °C in a humidified atmosphere containing 5% CO_2_. Every 2 days, the cells were sub-cultured, and an exponential phase of cells was used throughout the experiments.

### 4.4. Detection of Nitric Oxide (NO) Production

RAW 264.7 cells (1 × 10^5^ cell mL^−1^) were inoculated onto a 24-well plate. Subsequently, after 24 h, the cells were pre-incubated for 1 h with various concentrations of the sample at 37 °C. An additional incubation was carried out for a further 24 h with LPS (1 μg mL^−1^) at 37 °C. The quantity of nitrate accumulated in the culture medium was measured as an indicator of NO production after the required incubation [53]. Around 100 μL of the cell culture medium was mixed with 100 μL of Griess reagent (0.1% naphthyl ethylenediamine dihydrochloride and 1% sulfanilamide in 2.5% phosphoric acid), and the mixture was incubated at room temperature for 10 min. Fresh culture medium was used as a blank in all experiments. The optical density was measured at 540 nm using an enzyme-linked immunosorbent assay (ELISA) microplate reader (Sunrise, Tecan Co., Ltd., Melbourne, Australia).

### 4.5. Lactate Dehydrogenase (LDH) Cytotoxicity Assay

RAW 264.7 cells (1.5 × 10^5^ cells mL^−1^) were inoculated onto a 96-well plate. After 16 h, the cells were pre-incubated for 1 h with various concentrations of the sample at 37 °C. Then, at the same temperature, the cells were further incubated for another 24 h with LPS (1 μg mL^−1^). After the incubation, the LDH level in the culture medium was determined using an LDH cytotoxicity detection kit (Promega, Madison, WI, USA) according to the manufacturer’s instructions. A volume of 50 μL of the reaction mixture was added to each well, and the incubation was carried out for 30 min at room temperature in the dark. Afterwards, 50 μL of stop solution was added to each well, and the optical density was measured at 490 nm using an ELISA microplate reader (Sunrise, Tecan Co., Ltd., Melbourne, Australia).

### 4.6. Detection of Pro-Inflammatory Cytokines, Tumor Necrosis Factor-α (TNF-α) Interleukin-1β (IL-1β), and Interleukin-6 (IL-6) Production

The effects of inhibition of the sample on the production of pro-inflammatory cytokines from LPS-stimulated RAW 264.7 cells were investigated following a previously described method [54]. RAW 264.7 cells (1 × 10^5^ cells mL^−1^) were pretreated with the sample for 2 h and subsequently treated with LPS (1μg mL^−1^) to allow for the production of pro-inflammatory cytokines for 24 h. Supernatants were used for the assay using an ELISA kit (R & D Systems, Minneapolis, MN, USA) according to the manufacturer’s instructions.

### 4.7. Detection of Prostaglandin –E_2_ (PGE_2_) Production

RAW 264.7 cells (1 × 10^5^ cells mL^−1^) were pretreated with the sample for 2 h and subsequently treated with LPS (1 μg mL^−1^) to allow for the production of pro-inflammatory cytokines for 24 h. The PGE_2_ levels in the culture medium were quantified using a competitive enzyme immunoassay kit (R & D Systems, Minneapolis, MN, USA) according to the manufacturer’s instructions. The release of PGE_2_ was measured relative to the control value.

### 4.8. Western Blot Analysis

RAW 264.7 cells (1 × 10^5^ cells mL^−1^) were pretreated for 16 h and subsequently treated with LPS (1 μg mL^−1^) in the presence or absence of the sample. After 24 h, the cells were harvested and washed twice with ice-cold phosphate-buffered saline (PBS), and the cell lysates were prepared with lysis buffer (50 mmol^−1^ Tris–HCl (pH 7.4), 150 mmol^−1^ NaCl, 1% Triton X-100, 0.1% SDS) and 1 mmolL^−1^ethylenediaminetetraacetic acid (EDTA), which was kept for 20 min on ice. Cell lysates were centrifuged at 14,000× *g* for 20 min at 4 °C. Afterwards, using a BCA^TM^ protein assay kit, the protein concentrations in the supernatants were measured. Cell lysates containing 30 µg of protein were subjected to electrophoresis by sodium dodecyl sulfate-polyacrylamide 12% gels (SDS-PAGE). Separated proteins were then transferred onto nitrocellulose membranes (Bio-Rad, Hercules, CA, USA). The membranes were incubated with a blocking solution (5% skim milk in Tris-buffered saline containing Tween-20) for 90 min at room temperature. Subsequently, the membrane was incubated with anti-mouse iNOS (1:1000; Calbiochem, La Jolla, CA, USA) and anti-mouse COX-2 (1:1000; BD BiosciencPharmingen, San Jose, CA, USA) overnight at room temperature. After washing, the blots were incubated with horseradish peroxidase-conjugated goat anti-mouse IgG secondary antibody (1:5000; Amersham Pharmacia Biotech, Little Chalfont, UK) for 90 min at room temperature. Following the final washing step, the membrane was incubated with an ECL substrate mixture (PerkinElmer, MA, USA). The membrane was then sandwiched between cellulose acetate sheets and taped inside a film cassette, removing any air bubbles and exposing the hyper film in the dark for variable exposure periods depending on the strength of the signals. The film was developed using the appropriate developing solution and fixative. Subsequently, the film was washed and air-dried.

### 4.9. Origin and Maintenance of Zebrafish

Adult zebrafish were obtained from a commercial dealer (Seoul Aquarium, Republic of Korea), and 15 fish were kept in a 3-L acrylic tank (Tetra GmgH D-49304 Melle Made in Germany) under the following conditions: 28.5 °C, with a 14/10 h light/dark cycle. The day before, breeding took place, where 1 female and 2 males were interbred. In the morning (the onset of light), embryos were obtained from natural spawning, and the collection of the embryos was completed within 30 min in Petri dishes (containing media).

### 4.10. Waterborne Exposure of Embryos to Samples and LPS

The embryos (*n* = 15) were transferred to individual wells of 12-well plates containing 900 μL embryo media from approximately 7 to 9 hpf, and samples were added to the wells. After the incubation period of 1 h, 10 μg mL^−1^ LPS solution was added to the embryos exposed to samples for up to 24 hpf. Then, the embryos were rinsed using fresh embryo media.

### 4.11. Estimation of Oxidative Stress-Induced Reactive Oxygen Species (ROS) Generation and Image Analysis

The ROS production of the zebrafish was analyzed using an oxidation-sensitive fluorescent probe dye, 2,7-dichloro-dihydro fluorescein diacetate (DCFH-DA). The DCFH-DA was deacetylated intracellularly by nonspecific esterase, which was further oxidized to the highly fluorescent compound dichlorofluorescein (DCF) in the presence of cellular peroxides [55], At 4 dpf, the zebrafish larvae were transferred to one well of a 24-well plate, treated with DCFH-DA solution (20 µg mL^−1^), and incubated for 1 h in the dark at 28.5 ± 1 °C. After incubation, the zebrafish larvae were rinsed with fresh embryo media and anesthetized by 2-phenoxy ethanol (1/500 dilution sigma) before observation and photographed under the microscope CoolSNAP-Pro color digital camera (Olympus, Tokyo, Japan). The fluorescence intensity of individual zebrafish larvae was quantified using the Image J application.

### 4.12. Estimation of Oxidative Stress-Induced Nitric Oxide (NO) Generation and Image Analysis

The NO production of zebrafish was analyzed using a fluorescent probe dye, diaminofluorophore 4-amino-5-methylamino-2′,7′-dichlorofluorescein diacetate (DAF-FM DA). The transformation of DAF-FM DA by NO in the presence of dioxygen generates highly fluorescent triazole derivatives [56]. At 4 dpf, the zebrafish larvae were transferred to one well of a 24-well plate, treated with DAF-FM DA solution (5 µM), and incubated for 2 h in the dark at 28.5 ± 1 °C. After incubation, the zebrafish larvae were rinsed with fresh embryo media, anesthetized by 2-phenoxy ethanol (1/500 dilution sigma) before observation, and photographed under the CoolSNAP-Pro color digital camera microscope (Olympus, Tokyo, Japan). The fluorescence intensity of individual zebrafish larvae was quantified using the Image J application.

### 4.13. Estimation of Oxidative Stress-Induced Cell Death and Image Analysis

Cell death was detected in live embryos using PI staining. PI is a membrane impartment and is generally excluded from viable cells. PI is commonly used for identifying dead cells in a population. At 4 dpf, the zebrafish larvae were transferred to one well of a 24-well plate, treated with PI solution (80 µg mL^−1^), and incubated for 30 min under the dark at 28.5 ± 1 °C. After incubation, the zebrafish larvae were rinsed by fresh embryo media, anesthetized with 2-phenoxy ethanol (1/500 dilution sigma) before observation, and photographed under the CoolSNAP-Pro color digital camera microscope (Olympus, Tokyo, Japan). The Image J application quantified the fluorescence intensity of individual zebrafish larvae.

### 4.14. Statistical Analysis

The data are expressed as the mean ± standard deviation of three determinations. The data were analyzed via a one-way analysis of variance, followed by Duncan’s multiple range test (DMRT). *p* values < 0.05 were considered significant.

## 5. Conclusions

The isolation and characterization of 24-methylcholesta-5(6), 22-diene-3β-ol (MCDO) from the cultured marine diatom *P. tricornutum* and the in vitro and in vivo anti-inflammatory effects were evaluated. Even though the sterol compound has been identified in many studies, the anti-inflammatory activities using in vitro and in vivo models remained to be determined. Therefore, the anti-inflammatory effects of MCDO on LPS-induced inflammatory mediators in RAW 264.7 cells were investigated. MCDO significantly decreased the levels of pro-inflammatory mediators, including iNOS, NO, COX-2, PGE_2_, and IL-1β cytokine in LPS-stimulated macrophage cells. Furthermore, MCDO acted as a strong inhibitor of ROS and NO levels, protecting against the toxicity induced by LPS in inflammatory zebrafish embryos. MCDO isolated from the cultured marine diatom *P. tricornutum* revealed profound anti-inflammatory effects both in vitro and in vivo, suggesting that this major sterol may be a potent anti-inflammatory agent that contains a good source of bioactive lipids for the nutraceutical/food supplement industry.

## Figures and Tables

**Figure 1 marinedrugs-21-00231-f001:**
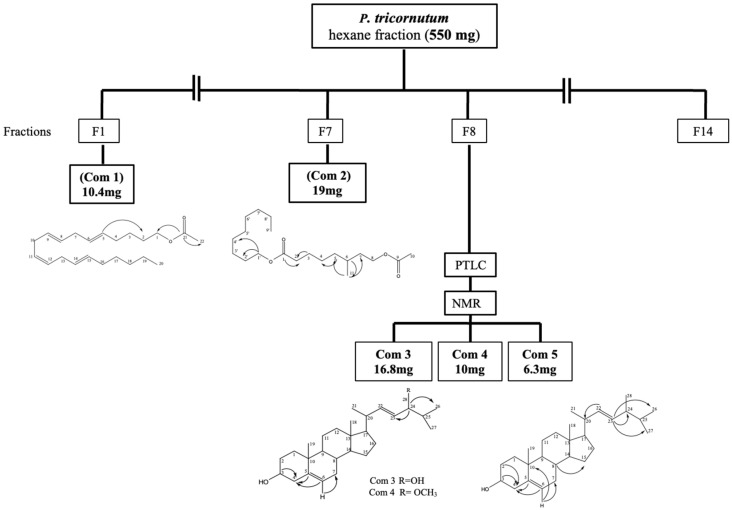
Extraction approaches followed by the bioactivity-guided fractionation from the cultured marine diatom *Phaeodactylum tricornutum* using sequential chromatographic and spectroscopic purification.

**Figure 2 marinedrugs-21-00231-f002:**
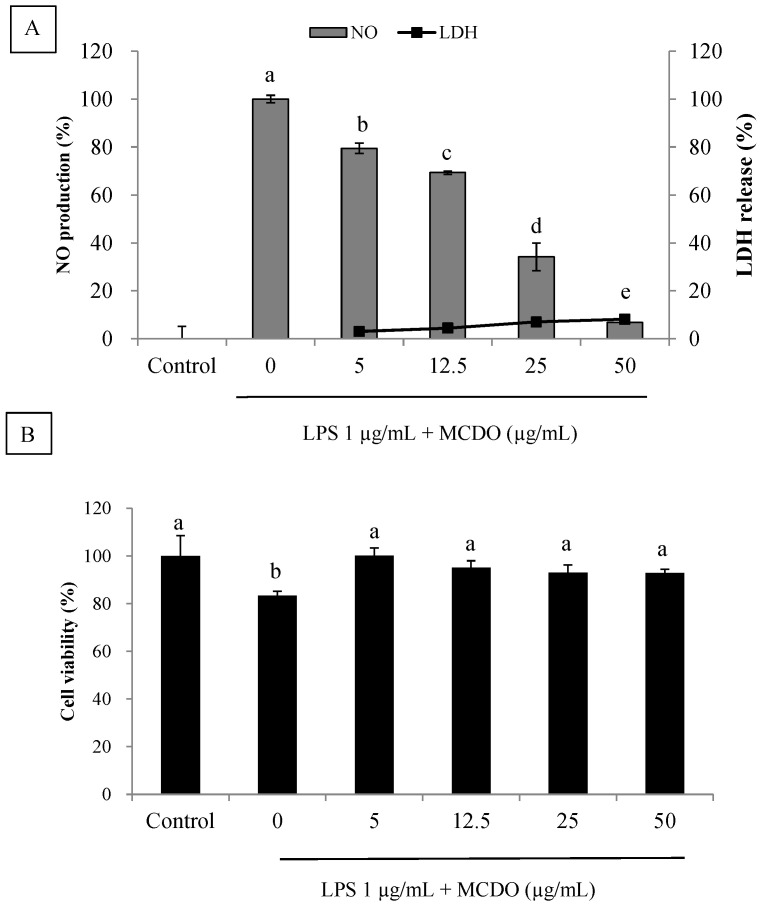
Inhibitory effects of MCDO on LPS-induced nitric oxide (NO) production and LDH release (**A**) and cell viability assay (**B**) in RAW 264.7 macrophages. Incubation of MCDO with cells for 24 h in response to LPS (1 μg mL^−1^) and NO levels in the medium were measured. Values with different alphabets (a, b, etc.) are significantly diverse at *p* < 0.05, as analyzed by DMRT. Values are expressed as means ± SD in triplicate experiments.

**Figure 3 marinedrugs-21-00231-f003:**
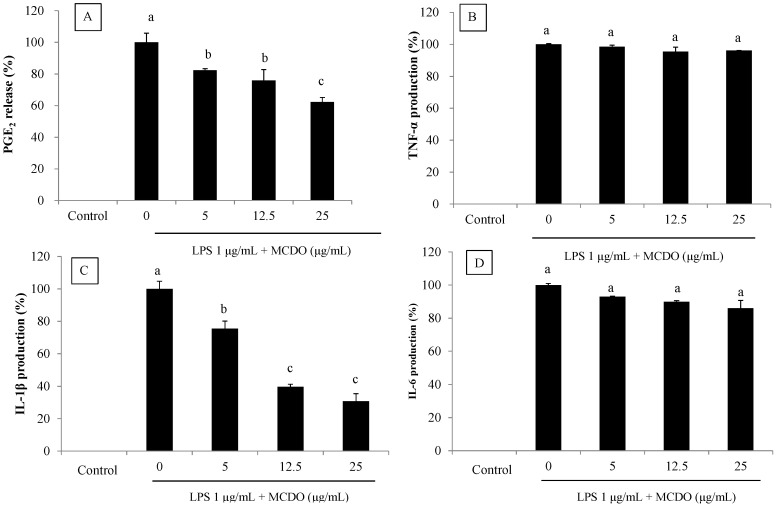
Inhibitory effects of MCDO on LPS-induced PGE_2_ (**A**), TNF-α (**B**), IL-1β (**C**), and IL-6 (**D**) production in RAW 264.7 macrophages. After incubation of cells with LPS (1 μg mL^−1^) for 24 h in the presence or absence of various concentrations of MCDO, the concentrations of PGE_2_, TNF-α, IL-1β, and IL-6 in the medium were measured. Values with different alphabets are significantly diverse at *p* < 0.05, as analyzed by DMRT. Values are expressed as means ± SD in triplicate experiments.

**Figure 4 marinedrugs-21-00231-f004:**
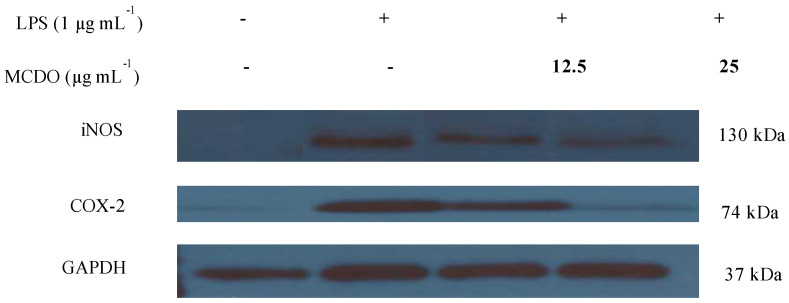
Inhibitory effects of MCDO on LPS-induced iNOS and COX-2 protein expression in RAW 264.7 macrophages. The cells were incubated with LPS for 24 h in the presence or absence of different concentrations of MCDO. Then cell lysates were subjected to electrophoresis, and the expression levels of iNOS and COX-2 were detected with specific antibodies.

**Figure 5 marinedrugs-21-00231-f005:**
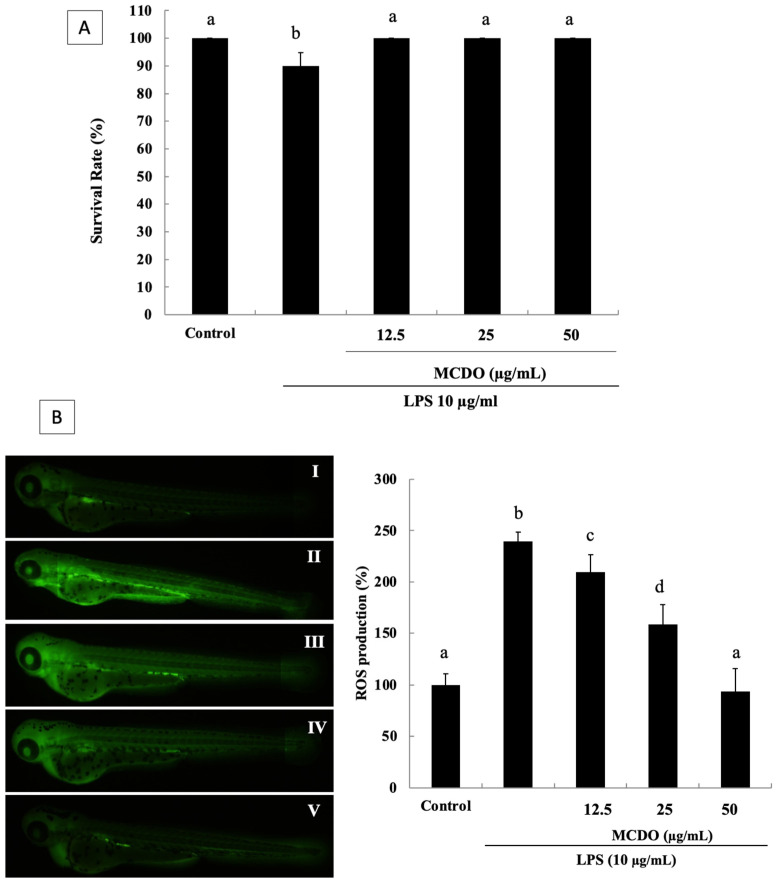
Survival rate after MCDO treatments at 12.5, 25, and 50 μg mL^−1^ concentrations against LPS-induced oxidative stress in zebrafish (**A**). Protective effects of MCDO on LPS-induced ROS level in zebrafish embryos, analyzed by DCFH-DA (**B**): Control (I); LPS 10 μg mL^−1^-treated (II); LPS 10 μg mL^−1^ + MCDO 12.5 μg mL^−1^-treated (III); LPS 10 μgmL^−1^ + MCDO 25 μg mL^−1^-treated (IV); and LPS 10 μg mL^−1^ + MCDO 50 μg mL^−1^-treated (V). Dexamethasone (50 μM) was used as a reference in this study. Experiments were performed in triplicate, and data are expressed as mean ± SE. Values with different letters (a, b, etc.) are significantly diverse at *p* < 0.05, as analyzed by DMRT.

**Figure 6 marinedrugs-21-00231-f006:**
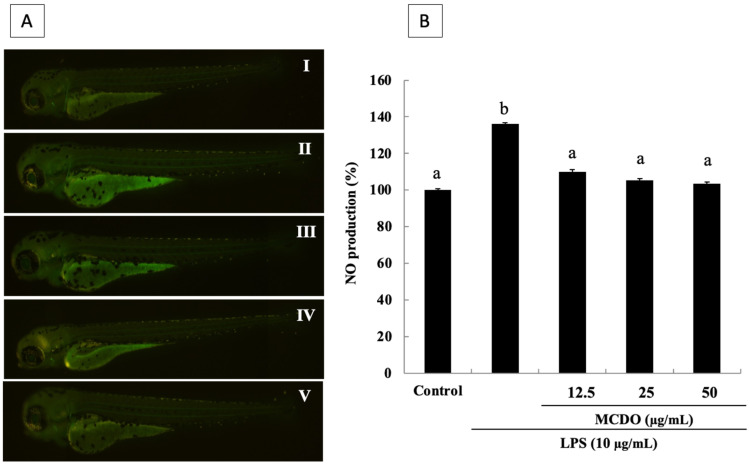
Protective effects of MCDO on LPS-induced nitric oxide (NO) production in zebrafish embryos (**B**), analyzed by DAF-FM DA (**A**): Control (I); LPS 10 μg mL^−1^-treated (II); (III) LPS 10 μg mL^−1^ + MCDO 12.5 μg mL^−1^-treated; LPS 10 μg mL^−1^ + MCDO 25 μg mL^−1^-treated (IV); and LPS 10 μg mL^−1^ + MCDO 50 μg mL^−1^-treated (V). Dexamethasone (50 μM) was used as a reference in this study. Experiments were performed in triplicate, and data are expressed as mean ± SE. Values with different letters (a, b) are significantly diverse at *p* < 0.05, as analyzed by DMRT.

**Figure 7 marinedrugs-21-00231-f007:**
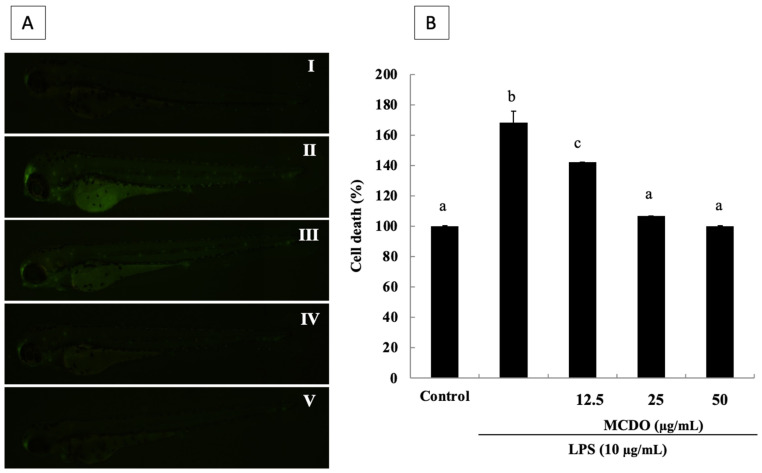
Protective effects of MCDO on oxidative stress-induced cell death by LPS in zebrafish embryos (**B**), visualized using PI staining (**A**): Control (I); LPS 10 μg mL^−1^-treated (II); LPS 10 μg mL^−1^ + MCDO 12.5 μg mL^−1^-treated (III); LPS 10 μg mL^−1^ + MCDO 25 μg mL^−1^-treated (IV); and LPS 10 μg mL^−1^ + MCDO 50 μg mL^−1^-treated (V). Dexamethasone (50 μM) was used as a reference in this study. Experiments were performed in triplicate, and data are expressed as mean ± SE. Values with different letters (a, b, etc.) are significantly diverse at *p* < 0.05, as analyzed by DMRT.

**Table 1 marinedrugs-21-00231-t001:** Molecular formula and IUPAC names of the isolated compounds from *Phaeodactylim tricornutum*.

Compound	Molecular Formula	IUPAC Name	Abbreviations
Com 1	C_22_H_36_O_2_	Icosa-5, 8, 11, 14-tetraenyl acetate	ITEA
Com 2	C_20_H_38_O_4_	Nonyl 8-acetoxy-6-methyloctanoate	NAMO
Com 3	C_27_H_44_O_2_	Cholestra-5(6), 22-diene-3, 24β-diol	CDDO
Com 4	C_28_H_46_O_2_	Cholestra-5(6), 22-dien-3, 24β-diol, methyl ether	CDDME
Com 5	C_28_H_46_O	24-methylcholesta-5(6), 22-diene-3β-ol	MCDO

**Table 2 marinedrugs-21-00231-t002:** NMR data for 24-methylcholesta-5(6), 22-diene-3β-ol (MCDO) in chloroform-D.

Position (C#)	δ_C_ ^a^ (ppm)	(mult)	δ_H_ ^a^ (mult, J_HH_ Hz)
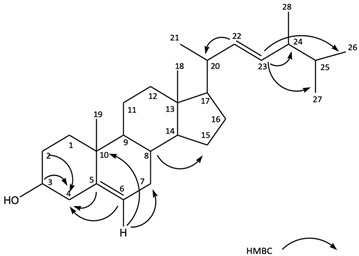			
1	37.32	(CH_2_)	1.85/1.02 m
2	31.73	(CH_2_)	1.50/1.42 m
3	71.9	(CH)	3.5 m
4	42.38	(CH_2_)	2.27/2.19 m
5	140.73	(C)	
6	121.71	(CH)	5.32 (1 H br d, J = 4.8 Hz)
7	31.98	(CH_2_)	1.98/1.85 m
8	43.15	(CH)	2.24 m
9	51.38	(CH)	0.89 m
10	36.5	(C)	
11	21.15	(CH_2_)	1.48/1.42 m
12	39.75	(CH_2_)	1.95/2.05 m
13	42.47	(C)	
14	56.85	(CH)	1.16 m
15	24.41	(CH_2_)	1.50/1.54 m
16	28.91	(CH_2_)	1.24/1.64 m
17	55.93	(CH)	1.14 m
18	12.13	(CH_3_)	0.68 s
19	19.49	(CH_3_)	0.99 s
20	40.37	(CH)	1.98 m
21	21.07	(CH_3_)	0.88 (3H d, J = 6.9 Hz)
22	136.63	(CH)	5.15 (1 H, dd, J = 8.4, J = 15.3 Hz)
23	131.63	(CH)	5.12 (1 H, dd, J = 8.4, J = 15.3 Hz)
24	50.13	(CH)	0.89 m
25	33.3	(CH)	1.48 m
26	20.24	(CH_3_)	0.89 (3 H, d, J = 6.3 Hz)
27	19.7	(CH_3_)	0.81 (3 H, d, J = 6.9 Hz)
28	18.78	(CH_3_)	0.99 (3 H, d, J = 7.0 Hz)

^a^ Multiplicity determined from HSQC-DEPT experiments.

## Data Availability

The data are contained within the article.

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
