# Peer review of "Structural Characterization and Anti-Inflammatory Effects of 24-Methylcholesta-5(6), 22-Diene-3β-ol from the Cultured Marine Diatom Phaeodactylum tricornutum; Attenuate Inflammatory Signaling Pathways"

_marinedrugs, 2023, doi:10.3390/md21040231_

Round 1

Reviewer 1 Report

The structural elucidation should be rewritten. The HMBC correlations must be redrawn.

Author Response

22nd March 2023

Dr. Kalpa W. Samarakoon

Institute for Combinatorial Advanced Research and Education (KDU-CARE),

General Sir John Kotelawala Defence University,

Kandawala Road, 10390,

Ratmalana, Sri Lanka.

The Editor-in-Chief/ Guest Editor,

Marine Drugs Editorial Office,

MDPI, Switzerland

Dear Sir/Madam,

Responses to the reviewers’ comments on the manuscript and resubmission of the revised manuscript

Manuscript ID: marinedrugs-2296380

Title: Structural characterization and anti-inflammatory effects of 24-methylcholesta-5(6), 22-diene-3β-ol from the cultured marine diatom, Phaeodactylum tricornutum; attenuate inflammatory signaling pathways

I would very much appreciate your valuable comments and suggestions provided to improve the quality of this manuscript. We hope the following responses and the corresponding revision of the manuscript fulfills the editor’s and reviewers’ requirements for considering this manuscript for publication in the Marine drugs, MDPI. The revised corrections and changes in the manuscript are highlighted in red color accordingly.

In the following section, we have shown the responses to each of the comments by the reviewer

Reviewer #2:

Comment 1 The structural elucidation should be rewritten. The HMBC correlations must be redrawn.

Responses: Following your comment, we have amended Table 2 including a new chemical drawing of the 24-methylcholesta-5(6), 22-diene-3β-ol (MCDO). The HMBC correlation was redrawn and corrections were made available to Table 2.

All the other amendments were done in the manuscript as required by reviewers. Moreover, I hope the new version of the manuscript is fulfilled with the reviewer’s comments and suggestions for publishing in the Marine drugs.

Thanking you in advance for your co-operation.

Kalpa W. Samarakoon (PhD)

(Corresponding author)

Reviewer 2 Report

In the manuscript (marinedrugs-2296380), authors identified the structure and evaluated the anti-inflammatory effects of 24-2 methylcholesta-5(6), 22-diene-3β-ol from the cultured marine diatom, Phaeodactylum tricornutum. Overall, the topic is of scientific interest and well-planned. I think the manuscript can be accepted and published in Marine Drugs after minor revision. Suggestions are provided below:

(1) Line 95: Authors are advised to give the yield of Com5, not just the mass.

(2) Figure 1-6: The meaning of the letters marked on the figures should be explained.

(3) This manuscript is the lack of positive control in the activity assay, and this makes it difficult to judge the activity of the prepared peptides. Therefore, the authors are advised to add positive controls to the activity test.

(4) 4.1 Materials: Please check carefully and add information on instruments or relevant reagents (e.g. immunoassay kit, ELISA kit). It is of utmost importance this is clarified and more detailed to allow replication.

(5) 4.2. Extraction, isolation, and characterization of compounds: It is suggested that the author make a figure to explain this content.

Author Response

22nd March 2023

The Editor-in-Chief/ Guest Editor,

Marine Drugs Editorial Office,

MDPI, Switzerland

Dear Sir/Madam,

Responses to the reviewers’ comments on the manuscript and resubmission of the revised manuscript

Manuscript ID:  marine drugs-2296380

Title: Structural characterization and anti-inflammatory effects of  24-methylcholesta-5(6), 22-diene-3β-ol from the cultured marine diatom,  Phaeodactylum tricornutum; attenuate inflammatory signaling pathways

I would very much appreciate your valuable comments and suggestions provided to improve the quality of this manuscript. We hope the following responses and the corresponding revision of the manuscript fulfills the editor’s and reviewers’ requirements for considering this manuscript for publication in the Marine drugs, MDPI. The revised corrections and changes in the manuscript are highlighted in red color accordingly.

Reviewer #1:

In the manuscript (marine drugs-2296380), authors identified the structure and evaluated the anti-inflammatory effects of 24-2 methylcholesta-5(6), 22-diene-3β-ol from the cultured marine diatom, Phaeodactylum tricornutum. Overall, the topic is of scientific interest and well-planned. I think the manuscript can be accepted and published in Marine Drugs after minor revision. Suggestions are provided below:

Comment 1: Line 95: Authors are advised to give the yield of Com5, not just the mass

Responses: Following your comment, we have included an additional table (Table 1) with 5 different compounds isolated from the cultured marine diatom, Phaeodactylum tricornutum.  The weights of each compound isolated were also added to the manuscript. The yield % of the Com 5 is included (Page 4; lines 117-118).

Comment 2: Figure 1-6: The meaning of the letters marked on the figures should be explained.

Responses: Agreed. According to the reviewer's comments, we have amended the figures' labeling and notations, and the ambiguity of data present in the text was corrected appropriately.

Comment 3: This manuscript is the lack of positive control in the activity assay, and this makes it difficult to judge the activity of the prepared peptides. Therefore, the authors are advised to add positive controls to the activity test

Responses:  Agreed.  While we conduct the in vitro inflammatory activity, Dexamethasone (50 μM) was used as a reference in this study. All the results were interpreted compared to the reference drug.

Comment 4:  Materials: Please check carefully and add information on instruments or relevant reagents (e.g. immunoassay kit, ELISA kit). It is of utmost importance this is clarified and more detailed to allow replication.

Responses:  Based on reviewers' suggestions, we have revised the manuscript and included relevant details (Manufacture, brand, model, and country) of the instruments used. 

Comment 5:  Extraction, isolation, and characterization of compounds: It is suggested that the author make a figure to explain this content.

Responses:  Thank you very much for the comment. According to the suggestion, a new figure (Figure 1) including a flow chart of extraction, separation, purification, and isolation of the compounds from the cultured marine diatom, Phaeodactylum tricornutum is added to the manuscript. page 3; (lines 95-101) and page 4; (lines 107-111).

Thanking you in advance for your co-operation.

Kalpa W. Samarakoon (PhD)

(Corresponding author)

Institute for Combinatorial Advanced Research and Education (KDU-CARE),

General Sir John Kotelawala Defence University,

Kandawala Road, 10390,

Ratmalana, Sri Lanka.
